# Developmental Vitamin D Deficiency in the Rat Impairs Recognition Memory, but Has No Effect on Social Approach or Hedonia

**DOI:** 10.3390/nu11112713

**Published:** 2019-11-08

**Authors:** Kathie Overeem, Suzy Alexander, Thomas H. J. Burne, Pauline Ko, Darryl W. Eyles

**Affiliations:** 1Queensland Brain Institute, The University of Queensland, St Lucia, QLD 4072, Australia; kathie.overeem@gmail.com (K.O.); suzy.alexander@uq.edu.au (S.A.); t.burne@uq.edu.au (T.H.J.B.); p.ko@uq.edu.au (P.K.); 2Queensland Center for Mental Health Research, The Park Center for Mental Health, Richlands, QLD 4076, Australia

**Keywords:** vitamin D, behavior, schizophrenia, developmental window, cognition

## Abstract

Developmental vitamin D (DVD) deficiency is a risk factor for schizophrenia. In rodents we show that DVD-deficiency alters brain development and produces behavioral phenotypes in the offspring of relevance to the positive symptoms of schizophrenia. The aims of this study are to examine behavioral phenotypes specific to the cognitive and negative symptoms of schizophrenia in this model, and to vary the duration of vitamin D deficiency during gestation and beyond birth. We hypothesize that a longer duration of DVD-deficiency would result in greater behavioral impairments. Female vitamin D-deficient Sprague Dawley dams were mated at 10 weeks of age. Dietary vitamin D was reintroduced to dams and/or pups at different developmental time-points: Conception, Birth, Post-natal day (PND) 6 and PND21. Adult male and female offspring were assessed on a battery of behavioral tests, including sucrose preference, open field, novel object recognition (NOR), social approach and social novelty. We find that all windows of DVD-deficiency impaired NOR a cognitive measure that requires intact recognition memory. Sucrose consumption, social approach and social memory negative symptom-like phenotypes were unaffected by any maternal dietary manipulation. In addition, contrary to our hypothesis, we find that rats in the Conception group, that is the shortest duration of vitamin D deficiency, demonstrate increased locomotor activity, and decreased interaction time with novel objects. These findings have implications for the increasing number of studies examining the preclinical consequences of maternal vitamin D deficiency, and continue to suggest that adequate levels of maternal vitamin D are required for normal brain development.

## 1. Introduction

Schizophrenia is a complex polygenic brain disorder with developmental origins [1]. Although symptom onset is usually delayed until the second or third decade of life, convergent evidence from the fields of epidemiology, imaging and post-mortem analysis strongly suggest that disruptions to foetal and early post-natal brain development contribute to an overall risk of schizophrenia [2]. With regards to epidemiological factors, studies link a diverse range of pre- and peri-natal environmental risk factors with schizophrenia [1,3,4]. In line with this, we establish that maternal vitamin D-deficiency is associated with an increased risk for schizophrenia in children [5,6]. 

In humans, vitamin D is largely obtained from cutaneous exposure to ultraviolet B radiation (UVB) from sunlight, which ultimately ends with the synthesis of the bioactive hormone vitamin D, 1,25-dihydroxyvitamin D3, although vitamin D can also be obtained from some dietary sources [7]. 

As a result, the average serum concentration of vitamin D follows a seasonal cyclic pattern with a 25% variation throughout the year [8]. We have confirmed that vitamin D levels are lowest in individuals born in late winter/early spring, while peak serum levels are observed following summer in early autumn [5]. Importantly, serum vitamin D concentrations are not totally agonal to the seasonal period. This is because vitamin D is stored in adipose tissue, which acts as a reservoir during depletion and repletion [9,10], serving to prolong levels after summer. During pregnancy there is increased demand on maternal stores of vitamin D, in part because of increased calcium transfer during the third trimester for fetal bone development [11,12,13]. Vitamin D deficiency and insufficiency is common in pregnant women, even during summer months [14,15]. Because of the seasonal variations in vitamin D levels, it is probable that different durations, or windows, of vitamin D deficiency will be experienced by the developing fetus. These windows could, in turn, impact risk for a variety of vitamin D-related conditions, including schizophrenia [16]. 

A rodent model of gestational developmental vitamin D (DVD)-deficiency has been developed in order to understand the neurobiology of vitamin D deficiency on brain development [17]. In the most published variant of the model, females are fed a diet without vitamin D for six weeks, resulting in the depletion of serum vitamin D. These females are then mated with vitamin D normal sires. Dams remain on this diet until birth, whereupon a normal vitamin D containing rat chow is reintroduced to the dam [18]. However, there are a number of studies that have examined the effect of varying the timeframe of vitamin D deficiency on the physiological and behavioral characteristics of the offspring. For example, we have shown an increase in ventricular size in DVD-deficient rats [17], which is increased in proportion to the duration of gestational vitamin D deficiency [19]. This is of interest, given ventriculomegaly is frequently observed in schizophrenia [20]. With regards to behavior, to date, the effect of DVD-deficiency on positive symptom phenotypes in adult offspring was thoroughly examined in the classic (gestational deficiency) DVD-deficiency model [21,22,23,24]. In addition, we show that varying the timeframe of vitamin D deficiency alters the expression of MK-801 induced hyperlocomoter activity. More specifically, when the period of maternal vitamin D deficiency is reduced by reintroducing dietary vitamin D at conception, this abolishes the well-described MK-801 sensitivity in the adult offspring [24]. Overall, this past research suggests that the timeframe of gestational vitamin D deficiency can affect both neuroanatomy and positive symptom-like phenotypes in the rat. Whether this is the case for cognitive and negative behavioral phenotypes is currently unknown. 

Although positive symptom phenotypes are repeatedly examined in this model, there is far less investigation of the so-called negative and cognitive symptom phenotypes. Our choice of test was governed by its ability to be relatively high throughput. This is why we choose novel object recognition and social memory recognition to assess cognitive behaviors and open field behavior, sucrose preference and social approach as tests of negative symptoms. We specifically choose widely used tests that allow given previous findings, showing that the duration of DVD-deficiency affects certain behavioral outcomes. Here we investigate the effects of different developmental windows of vitamin D deficiency on negative and cognitive symptom phenotypes. The overarching hypothesis is that a longer duration of vitamin D deficiency during gestation and into development would result in a greater behavioral impairment across all or some of these behavioral tasks. 

## 2. Materials and Methods

### 2.1. Animals

All procedures were performed with approval from The University of Queensland Animal Ethics Committee (QBI/404/14/NHMRC).

#### 2.1.1. Breeding

Female (*n* = 21) and Male (*n* = 15) Sprague Dawley rats were obtained from ARC, Western Australia. Female animals arrived at the facility at four weeks of age. They were housed in groups of four in standard polypropylene cages with high wire-top lids (54 × 36 × 30 cm). 

Vitamin D normal sires were single housed in smaller polypropylene cages with wire top lids (41 × 28 × 24 cm), and having free access to normal chow and water. Females were given one week to acclimate before being transferred to the experimental diet (Specially Feeds, Glen Forrest, Western Australia, Australia Cat # SF09-104). Control females remained on a diet containing 1 IU cholecalciferol/g, which is standard in most rat chows (Specialty Feeds, Glen Forrest, Western Australia, Australia Cat # SF09-105). 

Females remained on their diet for six weeks, a timeframe we repeatedly show in female rats to be adequate to deplete sera 25 hydroxy vitamin D (25OHD) levels to below assay sensitivity (2–4 nm prior to mating [17,18]. A limitation of this study is that we did not directly assess 25OHD levels in each dam. We did not obtain a blood sample from dams used in this study, as we wanted to avoid the additional stress this would place each dam under, possibly leading to off-target effects on dam–pup behavior, which may compromise the study. Although this treatment is sufficient to elevate parathyroid hormone (PTH) levels across all stages of pregnancy, dams and pups retain normal calcium and phosphate levels even if vitamin D depletion is prolonged until weaning [19]. After females were on the diet for six weeks, a sire was introduced to each box of four dams, with the day of introduction being considered the day of conception.

Recent modeling work based on developmental allometry of the brain across mammalian species was used to determine timeframes of gestational vitamin D deficiency in the rat, that correspond to known developmental timeframes in the human [25]. We reintroduced vitamin D normal diets at four different time-points: Conception, birth, PND 6, and weaning at PND21 (Figure 1). These time points approximate conception; post conception (PC) day 118 (second trimester); PC day 204 (third trimester); and PC day 535, or approximately nine months of age in human neurological development. 

#### 2.1.2. Offspring and Behavioral Testing

One hundred and eighty-eight male and female offspring were used. Same sex siblings were pair-housed in standard polypropylene cages with wire top lids (41 × 28 × 24 cm), kept on a constant 12h light/dark cycle (light phase 0600 h–1800 h), and provided with standard rat chow (Specialty Feeds, Glen Forrest, Western Australia, Australia) and water ad libitum. Animals remained in these housing conditionings over the duration of the behavioral testing, unless otherwise stated. All testing was conducted during the light phase. Furthermore, all behavioral testing, except for sucrose preference, was conducted in a room separate from the home housing room. Both the home-room and the behavioral testing room were located in the same animal facility with similar environmental conditions (room temp. 24 °C, 40–60% humidity). 

Prior to testing the animals were habituated to the behavioral room for 30 min. The behavioral room was maintained at low light conditions (2 × 50 W infrared bulbs and ambient light from a computer monitor). Testing began when the animals were approximately 100 days old. All animals were subjected to behavioral tests in the following order: Sucrose preference; open field; novel object recognition (NOR); social approach and social novelty. Testing was conducted over approximately 4 weeks.

### 2.2. Apparatus

Locomotion in the open field, NOR, and social approach and social novelty were conducted in three black opaque Perspex chambers. Chambers were 120 cm long, 60 cm wide, 60 cm high with a black brushed Perspex floor. For both open field and NOR assessment, a black Perspex insert was used to divide the chambers into two arenas measuring 60 × 60 × 60 cm. For assessment of social approach and social novelty, the central divider was removed and the chambers converted into 3-chamber apparatus using two clear Perspex dividers with three equal 40 × 60 × 60 cm compartments. The dividers had a square window (130 mm × 130 mm) in the middle of the bottom edge for rodent transition between chambers. Conspecifics for social approach and social novelty tests were corralled in holding columns 60 cm high with 1 cm Perspex bars spaced 1 cm apart. 

Each test was recorded using a centrally-placed video camera (Brickcom WFB-130Ap-71 Wireless Indoor 1.3 Megapixel Fixed Box, PoE Cam, WA, USA). Activity in the open field, social approach, and social novelty tests were analyzed post testing using automated video tracking software EthoVision XT v11.5 (Noldus Information Technology, Wageningen, The Netherlands) using differencing subtraction for subject identification. Activity during NOR was assessed manually using Observer v5.0 (Noldus Information Technology, Wageningen, The Netherlands) or the Manual Event Recorder module in EthoVision XT v11.5.

### 2.3. Sucrose Preference

Sucrose preference was assessed in the home room. For the duration of the 48 h test, rats were transferred to individual housing in polypropylene cages with wire top lids (41 × 28 × 24 cm). During testing, rats had access to a 1% sucrose solution and water in two 50 mL bottles connected to ball-bearing-regulated water sippers, the same sipper used on their standard water bottles. The side of water bottle placement was counterbalanced across cages. Consumption of water and sucrose was measured by weight at 24 h and 48 h. When measured at 24 h, the water and sucrose solution was replenished, and the position of the bottles reversed to eliminate side biases. Upon completion of the test, animals were returned to their previous housing configurations. 

### 2.4. Open Field

Rats were placed in the middle of the chamber and given 30 min to explore the novel environment. Six animals were examined simultaneously in separate chambers with diet group counterbalanced across chamber position in the behavioral room. After testing, the chamber floors and walls were cleaned with diluted detergent followed by a 70% ethanol solution, and then dried. 

### 2.5. Novel Object Recognition

Animals were habituated to an empty testing chamber for 10 min, then returned to their home cage, and the chamber was cleaned as described above. Two identical red and orange novel objects constructed of Duplo^®^ Lego^®^ were placed in the center of the chamber (120 cm apart) (Appendix A). Center placement of the object was chosen so that animals could walk freely around the edge of the arena, a habitual characteristic of animals in an open field. As a result, in order to interact with the objects, the animals were required to choose to move into the center of the chamber, rather than unintentionally encountering the object while exploring the arena perimeter. 

When animals were returned, they were placed in the center portion of the chamber with their heads positioned towards the wall so that their voluntary exploration with the objects was not biased by their placement in the chamber. Rats were given five minutes to freely explore the environment and objects. Interaction with objects was classified as nose touching. Animals were required to interact with each object for at least 2.5 s during the familiarization phase in order to be included in the subsequent memory test. Following the end of the five minute time-frame, animals were removed and returned to their home cage. In order to facilitate the use of visual rather than olfactory cues, the chamber and objects were thoroughly washed with diluted detergent followed by 70% ethanol. The inter-trial interval between habituation and testing was 1 h. Towards the end of the ITI the chambers were re-cleaned with a 70% ethanol solution, so that the smell of the initial learning phase was replicated for environmental consistency. One of the familiar objects encountered during familiarization was then placed in the chamber. A novel green object was also placed in the arena. It differed in configuration and color, but was approximately the same size (See Appendix A). The objects were placed in the center of the chamber in the same location as they were placed for familiarization. The position of the novel object (i.e., left or right) was counterbalanced across tests. Animals were re-introduced to the chamber in the same manner as during the habituation phase. The animals were given five minutes to freely explore the objects. 

### 2.6. Social Approach and Social Memory

#### 2.6.1. Conspecific Training

Age-matched conspecifics (30 females and 30 males) were used for this experiment. They were pair housed in the same home room as the test animals in wire top cages. Conspecifics were trained as described previously [26]. Briefly, the week before testing, thsse conspecifics underwent three daily training sessions in the experimental room. The conspecifics were given 30 min to acclimate to the room before training. During each training session, the animals were corralled in the vertical holding column for 10 min within the 3-chamber testing arena. Behavior was monitored for signs of distress (e.g., biting bars, trying to escape, excessive grooming). No animal showed signs of distress, and all quickly habituated to being placed in and removed from the holding columns. On testing days, conspecifics were transported from the home room to the experimental room with the test animals. Each conspecific was used twice per day. Once as conspecific 1, where they were used for both the social approach and social novelty test, and once as conspecific 2, where they were used as the novel rodent in the social novelty test. The same conspecifics were used over multiple testing days.

#### 2.6.2. Social Approach

Social approach and social memory were assed in a 3-chamber apparatus. Three animals were examined simultaneously with the group counterbalanced over the chamber position in the room. For habitation, animals were placed in the central chamber with their head facing towards a side wall, and not towards either chamber entrance, to help prevent a placement bias. They were given 10 min to freely explore and habituate to all three chambers and the empty holding columns. The animals were then removed and returned to their home cages while the first conspecific was corralled in a holding column. Of note, we did not clean the chambers between runs in order to maintain habituation to the arena. Furthermore, in order to delineate between actual social interaction over the drive to investigate a simple object, in the other chamber a Lego^®^ stimulus approximately the same size as a rat (but distinct from the object used in the NOR task) was placed in the opposite holding column. Position of the conspecific was counterbalanced across tests. Animals were placed in the 3-chamber apparatus as described above. They were given 10 min to freely explore all three chambers containing the new stimuli. They were then removed and placed back into their home cage.

#### 2.6.3. Social Novelty

The Lego^®^ stimulus was immediately replaced with a second conspecific rat, and the experimental rat returned to the testing apparatus in the same manner as described above. Again, animals were given 10 min to freely explore all chambers and stimuli. On completion animals were removed and the floor and walls of the chamber and holding columns thoroughly cleaned with diluted detergent followed by a 70% ethanol solution.

### 2.7. Data Analysis 

#### Overall Statistical Analysis

Our hypothesis was that behavioral phenotypes would be differentially impacted by the duration of developmental vitamin D (DVD)-deficiency. For each behavioral test, data was first quality checked for normality, and the effect of sex or sex-associated interactions using Analysis of Variance (ANOVA). Data from both sexes were combined if there were no sex-dependent effects. We were primarily concerned with identifying dietary groups that differ relative to controls. As outlined by Hsu (1999) the *F*-statistic is used to examine a null-hypotheses of homogeneity between all groups, and is consequently an incorrect “first-pass” criteria before direct comparisons relative to controls [27]. Thus, our primary analysis was a Dunnett’s-*t* style analysis with controls as the reference group, with a Bonferroni correction applied to the critical 0.05 *p*-value, resulting in a corrected alpha of <0.0125. Exceptions were tests where animals were required to make selections relative to chance, i.e., selection of sucrose solution over water, or interaction with a novel object over a familiar. In these tests results for each group were analyzed relative to a 0.50 chance threshold.

## 3. Results

### 3.1. Sucrose Preference

There was no difference between males and females for preference to sucrose (Appendix A). So data was examined pooled for sex. Analysis of total consumption of liquid (sucrose and water combined) did not reveal a significant difference for any group relative to controls (Figure 2a; Conception *p* = 1.00; Birth *p* = 0.53; PND06 *p* = 0.97; Weaning *p* = 0.75). Normality checks on the % preference data (ratio of sucrose solution consumed relative to the total consumption of both sucrose and water solution) revealed that the data was negatively skewed across all groups (Appendix A). This was driven by a number of animals who showed avoidance for the sucrose solution, a characteristic of some Sprague Dawley rats [28]. Analysis of animals classified as low sucrose consumers, here defined as animals with scores that differed significantly from their dietary group distribution, did not reveal a difference relative to the dietary group χ(4) = 2.44, *p* = 0.66. The data was therefore log10 transformed (Appendix A). Preference for sucrose relative to water was examined for each group by comparing group averages to the chance threshold (Figure 2b). All groups demonstrated a preference for sucrose at both the 24 and 48 h time points: (Control, 24 h *t*(37) = 6.5, *p* < 0.001, 48 h *t*(37) = 6.0, *p* < 0.001; Conception 24 h *t*(20) = 7.1, *p* < 0.001, 48 h *t*(22) = 5.4, *p* < 0.001; Birth 24 h *t*(23) = 4.3, *p* < 0.001, 48 h *t*(25) = 2.8, *p* = 0.01; PND06 24 h *t*(37) = 5.9, *p* < 0.001, 48 h *t*(37) = 4.7, *p* < 0.001; Weaning 24 h *t*(31) = 4.3, *p* < 0.001, 48 h *t*(31) = 6.0, *p* < 0.001). 

### 3.2. Open Field

There is no effect of sex on distance traveled, and therefore data is again examined pooled for sex. Previous studies show hyperlocomotor activity in a novel environment in DVD-deficient rats is most pronounced in the first few minutes of exposure [29]. Thus, the first five minute period of activity was examined. Analysis of the distance traveled over the first 5 min, expressed as one-minute time bins, revealed a significant increase in distance traveled by the Conception group relative to Controls *p* = 0.01 (Figure 3a). All other groups did not differ significantly (Birth *p* = 0.27, PND06 *p* = 0.99, and Weaning *p* = 0.06). Percent time spent in the center of the arena (50% of total arena size) did not vary between groups (*p* > 0.05) (Figure 3b). 

### 3.3. Novel Object Recognition

There was no effect of sex on distance traveled, and therefore data was again examined pooled for sex. Analysis of total object interaction time during familiarization revealed that the Conception group spent less time interacting with the objects relative to controls *p* = 0.005. All other groups did not differ from controls (Birth *p* = 0.23, PND06 *p* = 0.78, Weaning *p* = 0.051) (Figure 4a). During the familiarization phase, animals that did not interact with each object for a period of at least 2.5 s out of a total exploration time of at least 5 s, were removed from the subsequent recognition analysis (Table 1). Analysis of the proportion of excluded animals relative to dietary group using a Pearson’s Chi-square test confirms a difference between dietary groups, likely driven by the Conception group, *χ*(4) = 14.84, *p* = 0.005. 

One hour following the end of the familiarization phase, rats were placed back into the arena, where one familiar object had been replaced with a novel object. Rats were again given 5 min to explore both objects, and interaction was again defined as nose touching. The preference for a novel object only lasts 1–2 min, after which the preference diminishes as both objects become familiar and are explored equally. Thus, we analysed novelty preference only during the first min of exploration. Only the Control group exhibited normal preference for the novel object: Control *t*(41) = 2.8, *p* = 0.007; novel object recognition was impaired in all dietary conditions: Conception *t*(29) = 0.2, *p* = 0.88; Birth *t*(31) = 0.5, *p* = 0.65; PND06 *t*(21) = 0.7, *p* = 0.50; and Weaning *t*(32) = −1.5, *p* = 0.15 (Figure 4b).

### 3.4. Social Approach and Social Novelty

#### 3.4.1. Social Approach 

Unlike all other behaviors, there is a clear effect of sex on social behavior. Control males have intact social approach *t* = 3.8 *p* < 0.0005 and memory *t* = 4.3 *p* < 0.0001 when compared to a chance outcome. However control females do not display a preference for the conspecific *t* = 0.3 *p* > 0.05, although social memory is intact *t* = 3.2 *p* < 0.003. As a result, data for males and females were analyzed separately. All male offspring show a preference for the chamber containing the conspecific. However there is no effect of maternal diet when compared with dams on the control diet F (4–82) = 0.62 *p* > 0.05 (Figure 5a). In contrast, females display little preference for social interaction with percentage interactions for control females around chance i.e 51.3%. There are significant differences for social interaction between dietary groups; F (4–91) = 3.04 *p* < 0.05; however, no individual group differs from controls (Figure 5b).

In order to be included in the subsequent memory test it was necessary for all animals to be familiar with the conspecific. Thus, if they had not entered the chamber with the conspecific during the social approach phase of the test they were excluded for the memory test. Based on this criterion, a single male Control and female PND06 animal were removed from the analysis.

#### 3.4.2. Social Memory

In contrast to the selective effects of sex on social interaction, social preference for a novel conspecific is intact in both males and females. All male offspring show a preference for the chamber containing the novel conspecific. However there was no effect of maternal diet when compared with dams on the control diet F (4–83) = 1.10 *p* > 0.05 (Figure 5c). Similarly all female offspring show a preference for the chamber containing the novel conspecific. Again there is no effect of maternal diet when compared with dams on control diet F (4–91) = 0.97 *p* > 0.05 (Figure 5d). 

## 4. Discussion

The aims of this study were to establish whether negative and cognitive symptom phenotypes are present in DVD-deficient animals, and if so whether such phenotypes would be affected by duration of developmental deficiency. We predicted that longer durations of DVD-deficiency would result in greater impairments in behavior. We show here all that windows of DVD-deficiency produce small but nevertheless significant impairments in novel object recognition, a cognitive measure that requires intact recognition memory. Sucrose consumption, social approach and social memory, all of which are negative symptom-like phenotypes, are unaffected by maternal diet. In addition, contrary to our hypothesis, we find that animals born from vitamin D-deficient dams that are placed on normal vitamin D diets at conception, that is the shortest duration of gestational vitamin D deficiency, demonstrate increased locomotor activity in the open field and altered interaction time with novel objects. 

### 4.1. Object Recognition is Impaired Across All Durations of Maternal Vitamin D Deficiency

In the NOR paradigm, animals are required to recognize, or more specifically, separate familiar from unfamiliar stimuli [30]. This is demonstrated by virtue of increased interaction time with the novel object. To our surprise it would appear any developmental window of DVD-deficiency, no matter how short (i.e., reintroduction from conception), adversely affects this memory. 

A recent study in male Sprague-Dawley rats that expose offspring to vitamin D deficiency up until weaning confirms DVD-deficiency impairs NOR [31]. However, given recognition memory is also a component of social memory, we would have also expected to see deficits in this test, but this is not the case. We can only presume that the subtle impairments in memory recognition shown here to inanimate objects are partially compensated in a social setting by other cues such as ultrasonic vocalizations or olfaction. 

Our findings add to a growing list of previous studies in animals where gestational vitamin D deficiency also produces aberrant cognitive responding in the domains of memory and response-inhibition. More specifically, alterations are observed in the latent inhibition paradigm [32], the hole-board test [32], the 5-choice continuous performance task [33], the rodent gambling tasks [34] and the footshock-motivated brightness discrimination task [32]. The effects of varying developmental windows of vitamin D deficiency on these later cognitive behaviors is not yet assessed. 

### 4.2. No Effect of Vitamin D Deficiency on Primary and Secondary Negative Symptom Phenotypes

The sucrose preference and social approach paradigms are used to assess the impact of varying windows of maternal vitamin D deficiency on negative symptom phenotypes. Sucrose preference is driven by intact reward circuitry, comsummatory response and motivation to partake in behavior that eventuates in a reward response. All DVD-deficient animals exhibit intact sucrose preference. With regards to social preference, rats have an innate tendency for social approach, and show preference for time in the chamber with a novel conspecific. This test was originally designed for mice, but has also been adapted successfully to be used with the rat [35], including the Sprague Dawley strain [36]. Here we show males display normal social approach. Using the same criteria however, females show reduced preference for social interaction approaching chance outcomes. This is in line with past research showing that the propensity for social interaction is decreased in the female rat relative to the male [37,38]. Previous findings regarding social behavior in the 3-chamber apparatus in DVD-deficient rats are mixed. Although we have also shown alterations in social play behavior in DVD-deficient rats as juveniles, when using the 3-chamber apparatus, we again could find no social approach or memory deficit in rats deprived of vitamin D until weaning when assessed as adults [39]. This is in contrast to findings from another group who also deprived male rats of vitamin D until weaning, and who showed a small but significant reduction in social approach using the 3-chamber apparatus (Social memory was not tested) [31]. This variability in results is possibly due to experimental differences, such as different pre-habituation times, procedures regarding novel conspecific rotation and repeated vs no removal of the test animal during the three stages of this trial. Given social behaviors were tested last, it is also possible that prior repeated testing and handling abolishes any effect of DVD-deficiency. Indeed, this is exemplified by our previous work showing that prior handling and acute restraint to prior exposure in the open field abolishes all hyperlocomotor activity in DVD-deficient rats [29]. Taken together, our findings here do not suggest that DVD-deficiency in rats produces behaviors associated with the negative symptoms of schizophrenia.

Locomotor activity in the open field is used to assess anxiety and novelty-induced responses. Anxiety levels in our dietary groups are of particular interest, as this can influence the propensity to interact with stimuli, and could confound responses on NOR. Our observation that maternal diet has no effect on time in the center of the chamber relative to the edge suggests anxiety levels are not affected by dietary manipulation. With respect to overall locomotor activity, we previously show that hyperlocomotion in response to novel environments in DVD-deficient rats that were from dams made vitamin D replete from birth [29,40]. However, in this current experiment we observed hyperlocomotor activity selectively in the conception group and not the birth group. These conflicting results may also be due to variation in experimental factors, including differences in the test arena configuration, stimuli within the arena and experimental lighting conditions. More specifically, the previously observed hyperlocomotor activity is reported in the hole-board test [40], and a rectangular open field where half of the animals are tested under low white-light conditions. Our test was conducted in a simple open field under less-aversive red-light conditions. 

Thus, the outcome on this test appears sensitive to experimental characteristics again exemplified by our previous work showing that acute restraint prior exposure in the open field abolishes all hyperlocomotor activity in DVD-deficient rats [29].

### 4.3. DVD-Deficient Offspring Selective Effect of Window of Deficiency

The conception group exhibits abnormal patterns of behavioral responding in both the open field and when exploring novel objects. The conception group has the shortest duration of gestational vitamin D deficiency. These results, therefore, do not support our original hypothesis of longer timeframes of gestational vitamin deficiency, resulting in more pronounced behavioral impairments. It is possible that initial compensatory mechanisms (e.g., upregulation of CYP27A1 to synthesize more of the active vitamin D hormone) may be selectively present in this early pregnancy group. It is possible that with the reintroduction of this neuroactive steroid during early pregnancy, such mechanisms are over-activated that are either not present, or as influential at later developmental stages. However why such a mechanism would selectively lead to alterations in exploration remains unknown.

## 5. Conclusions

Maternal vitamin D deficiency is linked with increased risk of onset of schizophrenia in adulthood [5,6]. Here we examine whether longer time frames of maternal vitamin D deficiency confers greater behavioral impairments of relevance to the cognitive and negative symptoms of schizophrenia. Contrary to our hypothesis, we demonstrate any window of DVD-deficiency is sufficient to impair cognition, specifically in recognition memory. Secondly, no developmental window of DVD-deficiency has any effect on negative symptom phenotypes, as measured by a change in preference for sucrose or a social companion. This study adds to an increasing number of preclinical studies suggesting that adequate levels of maternal vitamin D are required for normal brain development. 

## Figures and Tables

**Figure 1 nutrients-11-02713-f001:**
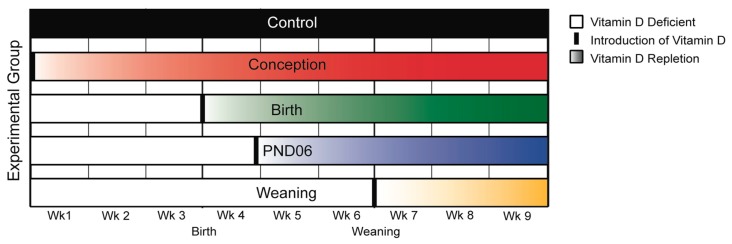
Dietary vitamin D reintroduction relative to group. The figure depicts when diets were changed from deplete (white) to control (colored) across groups. We have shown it requires 14 days for a vitamin D-deficient pregnant rat dam to achieve normal vitamin D levels once transferred to a control diet containing 1000 IU Cholecalciferol/kg [17,18]. The figure also depicts this gradual increase in serum 25OHD levels in offspring over 14 days following reintroduction of dietary vitamin D. Abreviations: Wk, week; PND Postnatal day.

**Figure 2 nutrients-11-02713-f002:**
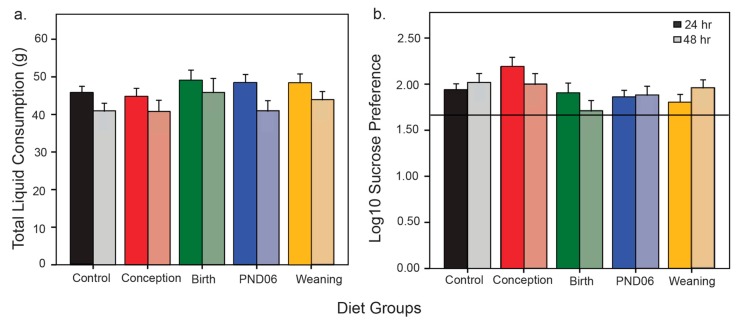
Assessment of preference for a 1% sucrose solution. Animals (Control *n* = 38; Conception *n* = 22; Birth *n* = 24; PND06 *n* = 38; Weaning *n* = 32) were single housed over the testing period. Results depict total liquid consumed during the 48 h testing period (**a**) and log10 converted sucrose preference relative to (log10 transformed) chance threshold (**b**) All groups were significantly above this threshold at both testing time-points.

**Figure 3 nutrients-11-02713-f003:**
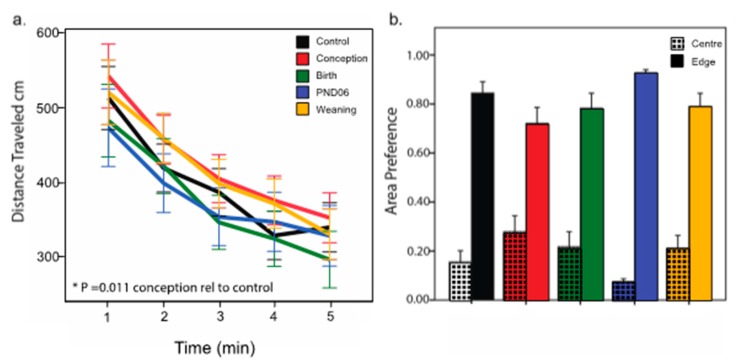
Assessment of locomotor activity in the open field. Animals (Control *n* = 42; Conception *n* = 41; Birth *n* = 32; PND06 *n* = 28; Weaning *n* = 40) were exposed to a novel 60 cm × 60 cm chamber in low light conditions. The results depict (**a**) distance traveled in one minute time bins over the 5-min testing period, and (**b**) time spent in the center of the chamber (50% of the total chamber size) relative to the edge. * *p* < 0.0125 (Bonferroni corrected).

**Figure 4 nutrients-11-02713-f004:**
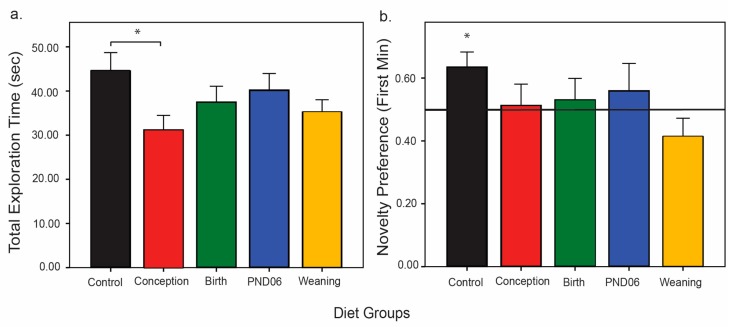
Assessment of novel object interaction and preference. Animals (Control *n* = 42; Conception *n* = 44; Birth *n* = 34; PND06 *n* = 28; Weaning *n* = 40) were given 5 min to interact with two identical novel objects. (**a**) Results depict interaction time over the entire familiarization period relative to group. (**b**) Only animals that explored each object for at least five seconds were included in the subsequent memory test 1-h post initial exploration. Results denote the animals (Control *n* = 42; Conception *n* = 32; Birth *n* = 32; PND06 *n* = 25; Weaning *n* = 37) preference for the novel over the familiar object relative to a 0.5 chance threshold. * *P* < 0.05.

**Figure 5 nutrients-11-02713-f005:**
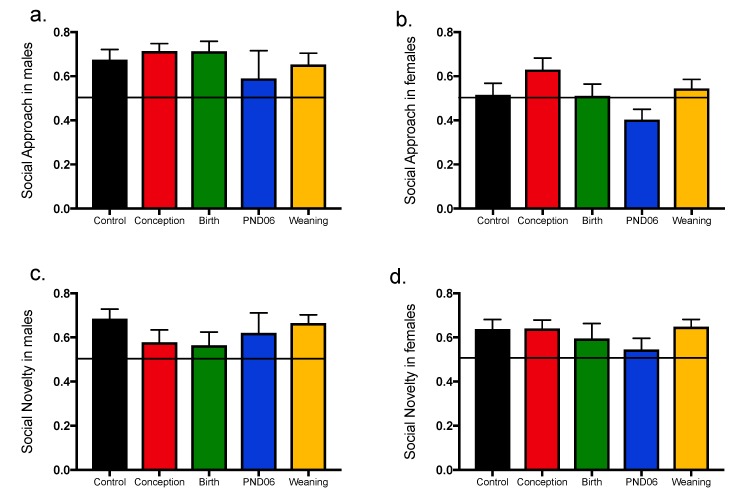
Assessment of social approach and social novelty in the 3-chamber apparatus. Animals (Control: Male *n* = 22, Female *n* = 20; Conception: Male *n* = 22, Female *n* = 21; Birth: Male *n* = 20, Female *n* = 14; PND06 Male *n* = 6, Female *n* = 20; Weaning: Male *n* = 18, Female *n* = 21). (**a**) Preference for the chamber containing the novel conspecific relative to a Lego^®^ stimulus by males with the 0.5 chance-threshold depicted. (**b**) Preference for the chamber containing the novel conspecific relative to a Lego^®^ stimulus by females with the 0.5 chance-threshold depicted. (**c**) Preference for a novel over a familiar conspecific during assessment of social novelty in males relative to a 0.5 chance threshold. (**d**) Preference for a novel over a familiar conspecific during assessment of social novelty in females relative to a 0.5 chance threshold.

**Table 1 nutrients-11-02713-t001:** Number of animals that were excluded from the novel object recognition analysis.

Diet Group *	Did Not Meet Criteria	Did Meet Criteria	Total
Control	1	41	42
Conception	12	32	44
Birth	3	31	34
PND06	3	25	28
Weaning	3	37	40
Total	22	188	88

Threshold for inclusion was set at 5 s interaction with each of the identical objects during the familiarization phase. * Significant difference observed using Pearson’s Chi-square test.

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
