# Peer review of "Developmental Vitamin D Deficiency in the Rat Impairs Recognition Memory, but Has No Effect on Social Approach or Hedonia"

_nutrients, 2019, doi:10.3390/nu11112713_

Round 1
Reviewer 1 Report
Thanks for the addition to the introduction. No further comments.
Author Response
Nothing further to address
Reviewer 2 Report
The revised manuscript addresses most of the concerns.
The discussion is missing a section on limitations and that should include the lack of actual lab data to verify historical findings of low levels in this model.
The authors note:
Females remained on diet for six weeks, a timeframe adequate to deplete sera 25 hydroxy vitamin D (25OHD) levels to below assay sensitivity (2-4 nM) prior to mating [17] [26]. Although this treatment is sufficient to elevate parathyroid hormone (PTH) levels across all stages of pregnancy, dams and pups retain normal calcium and phosphate levels even if vitamin D depletion is prolonged until weaning [19]. After females had been on diet for 6 weeks, a sire was introduced to each box of 4 dams.
However, they do not appear to have actually measured any of these in the present study. It is likely the findings are similar but we cannot know for certain. This should be included in the limitations along with any other points the authors consider to highlight as every study has multiple limitations
Author Response
please see attachment

This manuscript is a resubmission of an earlier submission. The following is a list of the peer review reports and author responses from that submission.
Round 1
Reviewer 1 Report
A well planned study and well written paper, but the introduction should provide further evidence that deficits in behavior, such as in cognition, can lead to schizophrenia. More importantly, the authors omit the possible role of infectious agents such as toxoplasma in the causation of human schizophrenia (pros and cons). This should be addressed in the introduction and discussion. The authors need discuss the evidence for schizophrenia being merely a developmental disorder in humans versus a disease caused by an infection or the inflammatory response to infection. Deficiency of vitamin D in utero or post gestation could lead to impaired immunity to toxoplasma or other agents in humans and hence the disease. The role of developmental disorders like cognition may not relate to the causation of human schizophrenia and there is no real evidence that the rats in the study exhibit the symptoms of human schizophrenia.
Reviewer 2 Report
The manuscript by Overeen and co-workers is very well written with clearly described methods and results. The conclusions drawn are supported by the data. The results are also, to the best of my understanding, comprehensively discussed.
My only comment relates to the introduction where the general reader might be helped by a brief passage that explains the rationale for why the authors chose to perform the tests of rodent behavior they did in this study.
Reviewer 3 Report
Thank you for the opportunity to review this manuscript entitled “Developmental Vitamin D deficiency in the rat impairs recognition memory but has no effect on social approach or hedonia”
This is an extremely well written manuscript that examines varyingduration of DVD-deficiency with vitamin D reintroduced to dams and/or pups at several different developmental time-points from conception to post-natal day 21. Adult male and female offspring were subsequently assessed on sucrose preference; open field; novel object recognition (NOR); social approach and social novelty. Unexpectedly the authors only found an impairment in NOR, a cognitive measures that requires intact recognition memory, but not in the other domains. However, their findings do reinforce other studies suggesting maternal vitamin D sufficiency is required for normal brain development.
Methods
Figure 1
Page 3 lines 107-109
The time points for humans are not clear. Conception is the same in both rats and humans, but not clear how birth in rats = second trimester in humans, etc.it seems second trimester in humans would still have certain pre-birth vulnerabilities as it relates to D deficiency that differ from those at birth.
Legends: The figure also depicts the gradual increase in serum 25OHD levels in offspring over 14 days following reintroduction of dietary vitamin D. see below
Results
No vitamin D levels are noted in the results section. Are the levels mentioned in figure 1 historical?
Are there data to validate D deficiency and D repletion?
Discussion:
Could the authors speculate more on why they believe DVD-deficiency produced small but nevertheless significant impairments in novel object recognition, a measure that requires intact recognition memory and why just this deficit?
While many of the findings are negative i think they are important and that message does not come through enough.